# Intravenous Fosfomycin for Difficult-to-Treat Infections: A Real-Life Multicentric Study in Italy

**DOI:** 10.3390/antibiotics14040401

**Published:** 2025-04-14

**Authors:** Verena Zerbato, Gianfranco Sanson, Lisa Fusaro, Valentina Gerussi, Sara Sincovich, Fabiana Dellai, Giovanni Del Fabro, Nicholas Geremia, Cristina Maurel, Donatella Giacomazzi, Chiara Biasinutto, Filippo Giorgio Di Girolamo, Gianfranco Scrivo, Venera Costantino, Manuela Di Santolo, Marina Busetti, Lory Saveria Crocè, Simone Giuliano, Massimo Crapis, George Zhanel, Carlo Tascini, Roberto Luzzati, Stefano Di Bella

**Affiliations:** 1Infectious Diseases Unit, Trieste University Hospital (ASUGI), 34125 Trieste, Italy; 2Clinical Department of Medical, Surgical and Health Sciences, Trieste University, 34129 Trieste, Italy; 3Infectious Diseases Clinic, Azienda Sanitaria Universitaria del Friuli Centrale (ASUFC), 33100 Udine, Italy; 4Department of Infectious Diseases, Santa Maria degli Angeli Hospital of Pordenone (AS FO), 33170 Pordenone, Italy; 5Unit of Infectious Diseases, Department of Clinical Medicine, Ospedale “dell’Angelo”, 30174 Venice, Italy; 6Unit of Infectious Diseases, Department of Clinical Medicine, Ospedale Civile “S.S. Giovanni e Paolo”, 30122 Venice, Italy; 7Pharmacy Unit, Santa Maria degli Angeli Hospital of Pordenone (AS FO), 33170 Pordenone, Italy; 8Pharmacy Unit, Trieste University Hospital (ASUGI), 34125 Trieste, Italy; 9Rehabilitation Unit, Trieste University Hospital (ASUGI), 34125 Trieste, Italy; 10Microbiology Unit, Trieste University Hospital (ASUGI), 34125 Trieste, Italy; 11Department of Medical Microbiology and Infectious Diseases, Max Rady College of Medicine, University of Manitoba, Winnipeg, MB R3E 0W2, Canada

**Keywords:** intravenous fosfomycin, difficult-to-treat infections, multidrug resistance

## Abstract

**Background:** Fosfomycin, an old antibiotic attracting renewed interest, offers a broad spectrum of activity and unique synergy with other agents. While widely used in severe infections, real-world data on intravenous fosfomycin remain limited. **Objectives:** This study aimed to describe the clinical and microbiological characteristics of patients treated with intravenous fosfomycin and to analyze its administration modalities in a real-world setting. **Methods:** A multicenter retrospective cohort study was conducted across five Italian hospitals. Adult patients receiving intravenous fosfomycin between January 2020 and December 2023 were included. **Results:** We enrolled 393 patients. The median age was 69 years, with most patients (45%) admitted to Critical Care Units. Pneumonia (34%), bloodstream infections (22%), and urinary tract infections (21%) were the most common indications. Gram-negative bacteria, particularly *E. coli* and *P. aeruginosa*, were the predominant pathogens. Fosfomycin was used as empirical therapy in 55% of cases and was combined with other agents in almost all cases (99%). The most frequent partners were piperacillin/tazobactam (21%) and new beta-lactam/beta-lactamase inhibitor combinations (18%). The median treatment duration was seven days, with most subjects (65%) receiving a fosfomycin dosage regimen of 16 g/day. Minimum inhibitory concentrations (MICs) values for fosfomycin were available for 61 isolates (15%), with 78.7% (48/61) showing MIC ≤ 32 mg/L. *C. difficile* infection occurred in only 2% of patients. Mortality rates at 30, 60, and 90 days were 21.6%, 26.7%, and 29.3%, respectively. **Conclusions:** This study provides valuable insights into the real-world use of intravenous fosfomycin.

## 1. Introduction

Fosfomycin, an antibiotic with a long-standing history in clinical practice since its discovery in 1969 [1], has recently attracted renewed interest in its intravenous disodium salt formulation for treating serious systemic infections, particularly those caused by multidrug-resistant (MDR) organisms. Among the existing antibacterial agents, fosfomycin has the lowest molecular weight (138 Da) and displays no structural relationship with other known classes of agents [2]. Upon entering the bacterial cell through permeases, fosfomycin covalently binds with UDP-N-acetylglucosamine enolpyruvyl transferase (MurA) and hampers its activity. Since this enzyme is crucial for the synthesis of N-acetylmuramic acid, a key precursor of peptidoglycan, its inhibition impedes bacterial cell wall formation, leading to cell lysis [3].

The low molecular weight of fosfomycin, along with its highly hydrophilic properties and negligible protein binding (only 3% of the drug is bound to serum proteins), results in favourable distribution throughout most tissues and interstitial fluid [4]. Studies have shown that the antibiotic achieves clinically relevant concentrations in serum, soft tissues, lung, bone, cerebrospinal fluid, and heart valves [1,5]. Furthermore, there is evidence of a high penetration rate into mature biofilms [6].

The common pathway for peptidoglycan layer construction in both Gram-positive and Gram-negative bacteria explains the broad spectrum of action exhibited by fosfomycin. Notably, among Gram-positive bacteria, it demonstrates considerable activity against *Staphylococcus aureus* (including methicillin-resistant *S. aureus*; MRSA), and various *Enterococcus* species, whether susceptible or resistant to glycopeptides. Regarding Gram-negative bacteria, the antibacterial spectrum encompasses most of the enteric Gram-negative bacteria, including strains producing extended-spectrum beta-lactamases (ESBL) and carbapenemases [7], although its efficacy against *Pseudomonas aeruginosa* is moderate [8]. Notably, the unique mechanism of action of fosfomycin prevents cross-resistance with other antibiotics [7].

A marked synergistic effect is observed when fosfomycin is combined with other antibacterial agents. Specifically, fosfomycin demonstrates complementary actions on cell wall synthesis stages when paired with β-lactams [9], while ciprofloxacin-induced membrane disruption enhances fosfomycin penetration [10]. Additionally, co-administration with aminoglycosides or glycopeptides has shown favourable results against *P. aeruginosa* and MRSA biofilms, respectively [11,12]. The synergistic effect of fosfomycin in association with the major antibiotic classes, has been explored in a recent systematic review [13], highlighting the potential of this partner drug in increasing bactericidal effect, preventing resistance, and pursuing a carbapenem- and colistin-spearing strategy.

The reference standard for fosfomycin antimicrobial susceptibility testing is agar dilution supplemented with glucose-6-phosphate. The susceptibility breakpoints for fosfomycin, as defined by the European Committee on Antimicrobial Susceptibility Testing (EUCAST), were recently revised to 8 mg/L. These breakpoints now apply exclusively to *Escherichia coli* infections of the urinary tract and to fosfomycin monotherapy [14].

With regard to adverse effects, fosfomycin can increase potassium renal excretion, leading to hypokalemia. Additionally, sodium overload may occur; however the antibiotic generally exhibits a good safety and tolerability profile [15].

All the previously addressed characteristics render fosfomycin suitable for various clinical applications. At present, its approval by the European Medicines Agency (EMA) is restricted to cases of serious infections where alternative antibiotic therapies are not feasible (bacterial meningitis, bone and joint infections, complicated intra-abdominal infections, complicated skin and soft tissue infections, complicated urinary tract infections (UTI), hospital-acquired pneumonia including ventilator-associated pneumonia, infective endocarditis, and bloodstream infection associated with any of the aforementioned patterns) [16]. There is currently no consensus on the dosing regimen. In clinical practice, daily doses usually range from 12 g to 24 g, divided into 2 to 4 administrations. Daily dose reduction is necessary for creatinine clearance of <40 mL/min [2]. Intravenous fosfomycin can be administered via continuous infusion and remains stable for at least 24 h in elastomeric pumps at both 4 °C and 34 °C [17].

Despite the widespread utilization of fosfomycin, limited data are available regarding its application in real-life scenarios [18,19,20,21]. The aims of this study were to (1) investigate the epidemiological and clinical characteristics of patients treated with intravenous fosfomycin; (2) investigate the pathogens isolated from patients treated with intravenous fosfomycin; and (3) investigate the administration modalities of intravenous fosfomycin.

## 2. Results

During the study period, a total of 393 patients were enrolled, with a median age of 69 years (IQR: 59–76), the majority of whom were male (n = 268, 68.2%).

At the time of fosfomycin administration, most patients were admitted to a Critical Care Unit (n = 178, 45.3%), while the remaining in Medical (n = 91, 23.2%), Surgical (n = 83, 21.1%) or Infectious Diseases (n = 41, 10.4%) wards.

The site of infection was identified in 84.5% of patients (n = 332). Overall, the most frequently affected sites were the lungs (n = 114, 34.3%), blood (n = 72, 21.7%) and urinary tract (n = 70, 21.1%), with a nearly equal prevalence for the latter two. However, when considering sites individually responsible for the infection (i.e., without other identifiable sources), the lungs remained the most prevalent site (n = 95, 28.6%), followed by urinary tract (n = 52, 15.7%) and the blood (n = 38; 11.4%). Other sites showed a prevalence of less than 10% (Table 1).

Microorganisms associated with the infection were identified in 243 patients (61.1%). In most cases (n = 195, 80.2%), the infection was monomicrobial. In the remaining cases, the infection was polymicrobial, involving two microorganisms in 43 patients (17.7%) and three microorganisms in 5 patients (2.1%). Among Gram-negative bacteria, *E. coli* was the most frequently detected infectious agent (n = 56, 23.0%), followed by *P. aeruginosa* (n = 55, 22.6%), and *Klebsiella pneumoniae* (n = 43, 17.7%). As regarding the resistance profile, fosfomycin was used to treat 32 extended-spectrum beta-lactamase (ESBL)-producing bacteria (more than half of which were *Escherichia coli*), 11 *K. pneumoniae* producing *K. pneumoniae* carbapenemase (KPC), 14 AmpC beta-lactamase producing bacteria, 7 OXA-48-producing *K. pneumoniae*, 6 extensively drug-resistant (XDR) *P. aeruginosa*, and 3 *Acinetobacter baumannii*. Only 2 metallo-beta-lactamase (MBL)-producing bacteria were reported, specifically 2 Verona integron-encoded metallo-beta-lactamase (VIM) producers (1 *Salmonella enterica* and 1 *P. aeruginosa*). Among Gram-positives, *S. aureus* (n = 32, 13.2%) was the most frequently detected, followed by *Enterococcus faecium* (n = 12, 4.9%), and *Staphylococcus epidermidis* (n = 12, 4.9%). As regarding the resistance profile, fosfomycin was used for 9 methicillin-resistant *S. epidermidis* (MRSE), 8 MRSA, and 7 *E. faecium* resistant to vancomycin (VRE).

The decision to administer antimicrobial drugs was based on an empirical strategy in nearly half of the cases (n = 214, 54.5%), while microbiological results guided the remaining prescriptions.

In the majority of subjects (n = 254, 64.6%), the fosfomycin dosage regimen was 16 g/day, followed by 12 g/day (n = 52, 13.2%), 24 g/day (n = 37, 9.4%), 2–8 g/day (n = 23, 5.9%), 18 g/day (n = 11, 2.8%), and 20 g/day (n = 9, 2.3%). In a small number of patients undergoing hemodialysis (n = 7, 1.8%), fosfomycin was administered at 2 g every 48 h. In 48.9% of cases (n = 192), fosfomycin was administered by continuous infusion. 

The median duration of fosfomycin therapy was 7 days (IQR 5–12). However, in one out of four cases, therapy was extended beyond 12 days, reaching up to a maximum of 120 and 144 days with continuous infusion in two subjects with osteomyelitis.

In all but three cases (0.8%), fosfomycin was co-administered with one (n = 287; 73.0%), two (n = 97; 24.7%), or three (n = 6; 1.5%) partner antibiotics. In most cases, these antibiotics were selected to act synergistically against the same pathogen, particularly in the treatment of multidrug-resistant Gram-negative infections. However, in a minority of patients, the co-administration may have targeted different pathogens in polymicrobial infections or concurrent infectious foci. The most frequently prescribed partner drug was piperacillin/tazobactam (n = 83; 21.1%), often used in combination with fosfomycin for infections caused ESBL-producing *Escherichia coli* or *Klebsiella pneumoniae*. New Beta-Lactamase Inhibitor Combinations (BLICs), such as ceftazidime/avibactam, ceftolozane/tazobactam, and meropenem/vaborbactam, were used in 72 cases (18.3%), mainly in the management of infections due to carbapenem-resistant Enterobacterales (CRE). Carbapenems (n = 70; 17.8%) and daptomycin (n = 42; 10.7%) were also frequently administered, with daptomycin primarily used for Gram-positive pathogens such as *Enterococcus faecium* or *Staphylococcus aureus*, including vancomycin-resistant strains (Figure 1).

For fosfomycin, MIC values were available for 61 isolates (15.3%), and in 44.3% of these cases, the isolates were *E. coli*. In 48 cases (78.7%), the MIC was ≤32 mg/L.

Only eight patients (2.0%) developed a *C. difficile* infection. The 30-, 60- and 90-day mortality rates were 21.6% (n = 85), 26.7% (n = 105), and 29.3% (n = 115), respectively.

## 3. Discussion

Intravenous fosfomycin is a promising drug that has been widely used in difficult-to-treat infections over the past decade. Although limited data are available regarding its use in real-world scenarios [18,19,20,21], the 2023 European Society of Cardiology (ESC) guidelines on infective endocarditis recommend it for the treatment of staphylococcal and enterococcal endocarditis [22]. The 2022 European Society of Clinical Microbiology and Infectious Diseases (ESCMID) guidelines on MDR infections also recommend its use in complicated UTI caused by third-generation cephalosporin-resistant Enterobacterales [23]. The Italian (SIMIT) and French (SPILF) Societies of Infectious Diseases recommend intravenous fosfomycin in combination with other drugs also for MBL and OXA-48 Gram-negatives producers [24].

In the present investigation, 393 patients were enrolled, most of whom were admitted to a Critical Care Unit. We provided an overview of intravenous fosfomycin use in a real-life scenario through a large dataset collected across five Italian hospitals. To our knowledge, this is the largest study providing real-life data on intravenous fosfomycin. The FORTRESS study is currently ongoing. This multicentric, prospective, non-interventional study, conducted in various European countries, aims to evaluate the efficacy and safety of intravenous fosfomycin for the treatment of several difficult-to-treat infections, and it is expected to provide additional real-life data on the use of intravenous fosfomycin [25].

In our study, fosfomycin was most commonly used for pneumonia (34.3%), bloodstream infections (BSI) (21.7%), and UTIs (21.1%). Other real-life studies have reported pneumonia and UTIs as the primary infectious sites for the use of intravenous fosfomycin [18,21]. With regard to the bacteria isolated, intravenous fosfomycin was mainly used to treat Gram-negatives, particularly *E. coli* and *P. aeruginosa*. Its use was also reported in infections caused by ESBL-producing pathogens. Among Gram-positive bacteria, the most commonly reported species were *S. aureus* (including MRSA) and *E. faecium* (including VRE). This reflects what has already been reported in other studies [18,19].

In the current investigation, fosfomycin was prescribed with another drug in almost all cases. The most common partner drugs were piperacillin/tazobactam, new BLICs, carbapenems, and daptomycin. Numerous in vitro and in vivo studies are available on the synergistic effects of intravenous fosfomycin in combination with other antibiotics [13]. Among these, we found piperacillin/tazobactam, new BLICs, and carbapenems, with varying percentages of synergism reported depending on the pathogen involved [13]. Daptomycin demonstrated a high level of synergism when combined with fosfomycin, particularly against *S. aureus* and *Enterococcus* species [26]. Our case series provides additional evidence of the ability of fosfomycin to synergize effectively with a broad spectrum of antibiotics, reinforcing its potential role in combination therapies against resistant pathogens. In our study fosfomycin was administered alone in only three cases. Some studies encourage the use of intravenous fosfomycin as monotherapy, at least in complicated urinary tract infections. The randomized clinical trial by Kaye et al. compared intravenous fosfomycin versus piperacillin/tazobactam in complicated UTIs, showing a similar clinical cure rate, both for ESBL and carbapenem-resistant *Enterobacterales* (CRE) infections [27]. In the FOREST trial, intravenous fosfomycin for bacteremic UTI caused by MDR *E. coli* showed comparable clinical and microbiological cure rates as compared to ceftriaxone/meropenem but failed non-inferiority criteria due to higher adverse event-related discontinuations [28]. Recently, a prospective, multinational matched-cohorts study on complicated UTIs caused by *E. coli* has been published, showing that targeted intravenous fosfomycin is a viable option in this setting [29].

Only 2% of patients developed *C. difficile* infection; however, this finding is difficult to attribute solely to fosfomycin, as it was almost always administered in combination therapy. The 30-day mortality rate was 21.6%, which is consistent with previously reported data in the literature [18,20]. This finding may be attributable to the severity of the infections treated in our population. In fact, nearly half of the cases were in Critical Care Units.

Our study has limitations that should be acknowledged. First, it is a retrospective study. Second, the patient population included was highly heterogeneous, which may limit the generalizability of the findings. Third, susceptibility testing for fosfomycin was not performed using the gold standard method, which could have affected the accuracy of the antimicrobial susceptibility results. Finally, clinical efficacy was not evaluated, as the study lacked a control group and the retrospective design made it challenging to assess treatment outcomes.

## 4. Materials and Methods

### 4.1. Study Design and Population

We conducted a multicenter retrospective observational study across five tertiary care hospitals in Italy. The study included all hospitalized adult patients (aged > 18 years) who received intravenous fosfomycin for at least 24 h during their hospital stay. The study period ranges from 1 January 2020 to 31 December 2023.

### 4.2. Data Collection and Definitions

The following data were retrospectively collected from hospital electronic medical records: demographics (age and gender); ward of hospital admission; details on intravenous fosfomycin administration (empiric or targeted use, dosage, duration, and any partner drugs); source of infection; isolated bacteria (including antibiogram results); date of death; and occurrence of *Clostridioides difficile* infection during hospital stay.

In all five institutions, intravenous fosfomycin is prescribed exclusively following consultation with an Infectious Diseases specialist. The date of first fosfomycin administration (considered as day-1) was collected. The length of fosfomycin therapy was computed as the difference between day-1 and the date of last administration. Moreover, the date of possible patient’s death was documented, and the mortality rate after 30, 60 and 90 days from day-1 was also calculated.

All isolates were identified by MALDI-TOF mass spectrometry (bioMérieux, Marcy-l’Etoile, France), and antimicrobial susceptibility, including minimum inhibitory concentrations (MICs) for fosfomycin, was assessed with the Vitek2 system (bioMérieux, Marcy-l’Etoile, France). The agar dilution method, considered the gold standard for fosfomycin MIC determination, was not employed.

All data were pseudonymized via a web-based central, password-protected clinical database management system.

### 4.3. Statistics

Continuous variables were presented as median and interquartile range (IQR), while the categorical variables as frequency and percentage.

### 4.4. Ethics

This study was performed in line with the principles of the Declaration of Helsinki. Approval was granted by the Ethics Committee of University of Trieste (Date 24 October 2023/N°V135). Data were pooled and collected anonymously.

## 5. Conclusions

Intravenous fosfomycin appears to be widely used in managing difficult-to-treat infections, particularly in critically ill patients. This study provides an overview of its application against both Gram-negative and Gram-positive pathogens, including MDR strains, with combination therapy being the most common approach. The most frequently treated infections were pneumonia, bloodstream infections, and urinary tract infections. While its real-world applicability is evident, the findings emphasize the need for further research to establish standardized dosing regimens and evaluate its potential as a monotherapy. Additionally, prospective studies are warranted to better assess fosfomycin’s clinical outcomes.

## Figures and Tables

**Figure 1 antibiotics-14-00401-f001:**
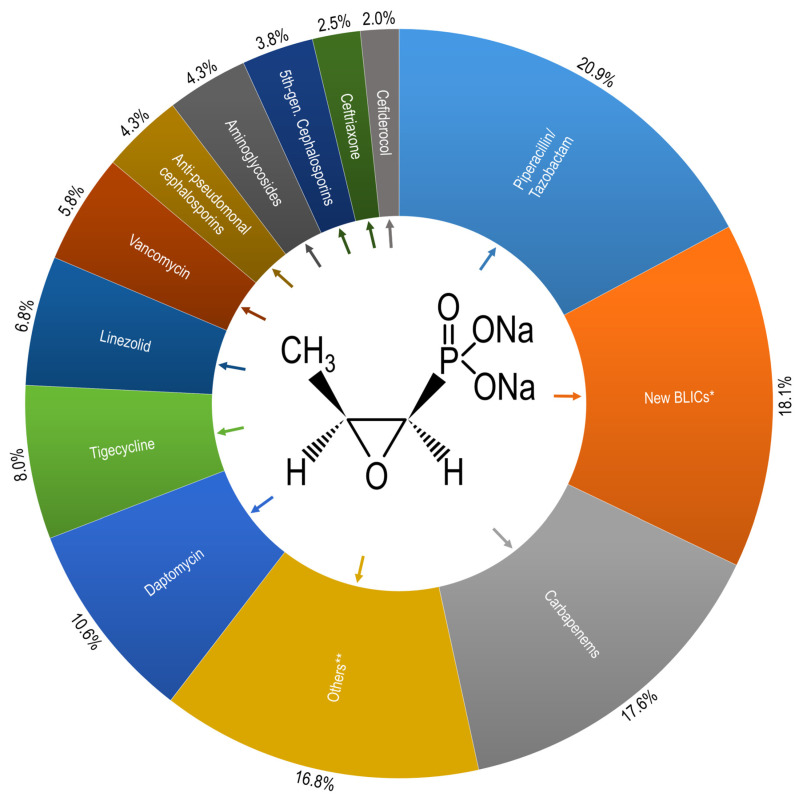
Frequency of partner antibiotics administered with intravenous fosfomycin. BLICs* = Beta-Lactamase Inhibitor Combinations, such as ceftazidime/avibactam, ceftolozane/tazobactam, and meropenem/vaborbactam; Others**: oxacillin, cefazolin, fluoroquinolones, oritavancin, dalbavancin, and colistin.

**Table 1 antibiotics-14-00401-t001:** Distribution on infection rates (percentages) according to the involved districts in 332 patients.

District	Blood	Bone	CNS	Genitals	Heart	Abdomen	Lung	Prosthetic Implants	Skin and Soft Tissue	Urinary Tract	Vascular System
Blood	11.4	1.2	0	0	0	0.6	3.0	0.3	0.6	4.2	0.3
Bone	1.2	6.3	0.3	0	0	0	0.3	0	0	0	0
CNS	0	0.3	0.9	0	0	0	0	0	0	0	0
Genitals	0	0	0	0.3	0	0	0	0	0.3	0.3	0
Heart	0	0	0	0	4.8	0	0.6	0	0	0	0
Abdomen	0.6	0	0	0	0	8.7	0.6	0	0	0	0.3
Lung	3.0	0.3	0	0	0.6	0.6	28.6	0	0.6	0.9	0.3
Prosthetic implants	0	0	0	0	0	0	0	2.7	0	0	0
Skin and soft tissue	0.6	0	0	0.3	0	0	0.6	0	5.1	0	0
Urinary tract	4.2	0	0	0.3	0	0	0.9	0	0	15.7	0
Vascular system	0.3	0	0	0	0	0.3	0.3	0	0	0	0.6

Data in cells at the intersection of a row and a column describing a same district (e.g., lung and lung) report the percentage of patients in which that district was recognized as the only source of infection, i.e., without other infections. Data at the intersection of different districts (e.g., lung and bloodstream) indicate the percentage of patients in which infections were identified in both sites. CNS: central nervous system.

## Data Availability

The datasets analyzed during the current study are not publicly available but are available from the corresponding author upon reasonable request.

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
