# Peer review of "Intravenous Fosfomycin for Difficult-to-Treat Infections: A Real-Life Multicentric Study in Italy"

_antibiotics, 2025, doi:10.3390/antibiotics14040401_

Round 1

Reviewer 1 Report

Comments and Suggestions for Authors

Fosfomycin is in high demand these days due to the increasing prevalence of drug-resistant infections. The authors have conducted this retrospective study to reflect real-life experiences with this medication, successfully gathering a large number of cases—a notable strength of the study. However, several aspects need revision and improvement for this article to serve as a valuable addition to the existing literature and to engage readers effectively. Below are some suggestions.

1)The flow of data in the introduction needs to be reconsidered and rearranged. While all the information presented appears relevant and important, the current structure causes the reader to jump between different points. Consider revising the background and organizing the information to improve coherence and readability. For example, the discussion on molecular weight and physicochemical properties in the fifth paragraph should be moved up to follow the first mention of molecular weight, ensuring a more logical progression of ideas.

2) The title refers to difficult-to-treat infections, but authors do not clarify whether it is referring to the internationally recognized definition or a different one. Unfortunately, the results include infections /organisms that do not align with the standard international criteria !!

3) It is unclear whether the antibiotics combined with fosfomycin were intended to target the same organism or if their administration was coincidental, addressing different infections or organisms. Additionally, including specific examples of the identified organisms and the antibiotic combinations used would be beneficial for the reader.

4) It is unclear why fosfomycin was chosen for organisms that typically have several other effective treatment options (e.g., ESBL, AmpC-producing bacteria). Clarifying the rationale behind its use in these cases—whether due to resistance patterns, patient-specific factors, or other considerations—would strengthen the study's clinical relevance and provide valuable insights for readers.

5)The first few sentences in the discussion should be removed, as they were intended as guidance for the authors and are not meant for publication

Author Response

Comments and Suggestions for Authors

Fosfomycin is in high demand these days due to the increasing prevalence of drug-resistant infections. The authors have conducted this retrospective study to reflect real-life experiences with this medication, successfully gathering a large number of cases—a notable strength of the study. However, several aspects need revision and improvement for this article to serve as a valuable addition to the existing literature and to engage readers effectively. Below are some suggestions.

1)The flow of data in the introduction needs to be reconsidered and rearranged. While all the information presented appears relevant and important, the current structure causes the reader to jump between different points. Consider revising the background and organizing the information to improve coherence and readability. For example, the discussion on molecular weight and physicochemical properties in the fifth paragraph should be moved up to follow the first mention of molecular weight, ensuring a more logical progression of ideas.

We thank the reviewers for their suggestion. We have made the requested changes.

2) The title refers to difficult-to-treat infections, but authors do not clarify whether it is referring to the internationally recognized definition or a different one. Unfortunately, the results include infections /organisms that do not align with the standard international criteria !!

We thank the reviewer for the valuable comment. We are aware that the term “difficult-to-treat” may, in some contexts, refer specifically to a microbiological definition involving Gram-negative pathogens resistant to all first-line agents (Kadri et al.). However, in our study, the term was used more broadly and intentionally, to reflect clinical scenarios where treatment is particularly challenging — due to pathogen resistance, but also site of infection, patient frailty, or limited treatment options. This choice was made to better represent the real-life heterogeneity of cases included in our multicentric experience. Given the nature of the study and its focus on intravenous fosfomycin use in such complex situations, we believe the title remains appropriate and consistent with the manuscript’s content.

3) It is unclear whether the antibiotics combined with fosfomycin were intended to target the same organism or if their administration was coincidental, addressing different infections or organisms. Additionally, including specific examples of the identified organisms and the antibiotic combinations used would be beneficial for the reader.

4) It is unclear why fosfomycin was chosen for organisms that typically have several other effective treatment options (e.g., ESBL, AmpC-producing bacteria). Clarifying the rationale behind its use in these cases—whether due to resistance patterns, patient-specific factors, or other considerations—would strengthen the study's clinical relevance and provide valuable insights for readers.

We thank the reviewers for their comments. We update the results as suggested.

5)The first few sentences in the discussion should be removed, as they were intended as guidance for the authors and are not meant for publication.

We thank the reviewers for their suggestion. We removed the first sentences of the discussion.

Reviewer 2 Report

Comments and Suggestions for Authors

In this manuscript, the authors detail the use of IV fosfomycin in treating MDR infections. Although well written, one of the biggest shortcomings of this paper is that the authors did not address the significance of this study/ findings. The correlation between the use of fosfomycin in the treatment regimen and patient outcomes needs to be included in this study.

Line 145: I am curious to know that a small percentage of patients who received 2g of fosfomycin every 48 hrs, what other drugs were specifically combined for this regimen, and what the outcome of using this dose was.

Line 233-237: This is redundant and does not fit under the "material and methods" section.

Please address the significance of this study and how it addresses the knowledge gap in the field of combination therapies that include fosfomycin. 

Line 166-167: Any hypothesis that explains the higher mortality rate at day 90?

Author Response

In this manuscript, the authors detail the use of IV fosfomycin in treating MDR infections. Although well written, one of the biggest shortcomings of this paper is that the authors did not address the significance of this study/ findings. The correlation between the use of fosfomycin in the treatment regimen and patient outcomes needs to be included in this study.

We thank the reviewer for this comment. However, we would like to clarify that the primary aim of our study was not to assess the efficacy of intravenous fosfomycin in terms of clinical outcomes, but rather to describe its use in real-life clinical practice. As stated in the manuscript, the objectives were to investigate the epidemiological and clinical characteristics of patients treated with IV fosfomycin, the pathogens involved, and the modalities of fosfomycin administration. While we did report patient outcomes for completeness, the study was not designed nor powered to evaluate the causal relationship between fosfomycin use and clinical outcomes.

Line 145: I am curious to know that a small percentage of patients who received 2g of fosfomycin every 48 hrs, what other drugs were specifically combined for this regimen, and what the outcome of using this dose was.

This dosage is the one recommended for dialysis. We have added this specification in the results.

Line 233-237: This is redundant and does not fit under the "material and methods" section.

We thank the reviewer for the suggestion. We have moved this section to the end of the introduction.

Please address the significance of this study and how it addresses the knowledge gap in the field of combination therapies that include fosfomycin. 

Thank you for the suggestion. We have added a sentence in the discussion about this topic. 

Line 166-167: Any hypothesis that explains the higher mortality rate at day 90?

We think that the high mortality rate is justified by the clinical setting in which the drug was used (nearly half of the cases were in intensive care units) and by the type of pathogens involved, many of which were multidrug-resistant. We thank you for the suggestion. We had already included this point in the discussion, but we have now reinforced the concept.

Reviewer 3 Report

Comments and Suggestions for Authors

Intravenous fosfomycin for difficult-to-treat infections: a real- life multicentric study in Italy by Zerbato et al. deals with use of fosfomycin in various infectious diseases in clinical settings. In general the paper is written nicely, although some changes are suggested for improving the manuscript:

  1. The title and abstract are fine, they give good overview of topic.
  2. Introduction is fine, all important aspects of fosfomycin use are covered.
  3. Results are fine, I appreciate showing both number and percent for each given info. I highly dislike the Figure 1 as it is done quite disproportional and non-aesthetically - please make changes, as this is only a matter of appearance. 
  4. Discussion section begins with text forgotten from template, please eliminate it.
  5. MM section - Why were not data on patients presented in more details? Sex, age, other comorbidities, use of other drugs, any other data that may affect the use and effectiveness of drugs, fosfomycin in this case. Minimum inhibitory concentrations (MICs) for 
    fosfomycin were not determined through agar dilution method----if they were not obtained by this method, how were they obtained, please explain?
  6. Conclusion is fine.

However, this issue I have with this paper is large number of Authors. I am aware of the study being clinical, and involving large number of patients, still, retrospective study, observational in character, with not so many details presented- should not have more than 10 Authors. I have personally published papers with interventional prospective design of the study, and have not passed 7 Authors. So, please consider reducing this number.

Best of wishes in publishing your paper!

Author Response

Intravenous fosfomycin for difficult-to-treat infections: a real- life multicentric study in Italy by Zerbato et al. deals with use of fosfomycin in various infectious diseases in clinical settings. In general the paper is written nicely, although some changes are suggested for improving the manuscript:

The title and abstract are fine, they give good overview of topic.

Introduction is fine, all important aspects of fosfomycin use are covered.

Results are fine, I appreciate showing both number and percent for each given info. I highly dislike the Figure 1 as it is done quite disproportional and non-aesthetically - please make changes, as this is only a matter of appearance. 

We appreciate the reviewer’s feedback regarding Figure 1. However, we believe that its current design effectively conveys the intended message and plays a key role in supporting the main arguments of the paper. For this reason, we kindly prefer to keep it as is.

Discussion section begins with text forgotten from template, please eliminate it.

We thank the reviewers for their suggestion. We removed the first sentences of the discussion.

MM section - Why were not data on patients presented in more details? Sex, age, other comorbidities, use of other drugs, any other data that may affect the use and effectiveness of drugs, fosfomycin in this case. 

We present the general data at the beginning of the Results section. The list of fosfomycin partner drugs is then detailed throughout the Results. Unfortunately, comorbidities were not collected.

Minimum inhibitory concentrations (MICs) for fosfomycin were not determined through agar dilution method----if they were not obtained by this method, how were they obtained, please explain?

We thank the reviewers. We have expanded this paragraph to provide more detailed information on this aspect.

Conclusion is fine.

However, this issue I have with this paper is large number of Authors. I am aware of the study being clinical, and involving large number of patients, still, retrospective study, observational in character, with not so many details presented- should not have more than 10 Authors. I have personally published papers with interventional prospective design of the study, and have not passed 7 Authors. So, please consider reducing this number.

Best of wishes in publishing your paper!

We thank the reviewer for this observation. We understand the concern regarding the number of authors. However, this study involved the collaboration of five different hospitals, and we felt it was important to acknowledge the contributions of all those who played a significant role in the data collection, analysis, and interpretation. Each author met the authorship criteria, and their involvement was essential for the realization of the study. We hope the reviewer will understand our intention to properly recognize the collaborative nature of this work.

Round 2

Reviewer 2 Report

Comments and Suggestions for Authors

The reviewers have addressed all the concerns! 

Author Response

Thank you for your kind help in improving the manuscript.

Reviewer 3 Report

Comments and Suggestions for Authors

The Authors have improved the manuscript, I believe it can be published now. Kindest regards.

Author Response

(The authors gave the same response as above.)
